# Securing the Cyber Resilience of a Blockchain-Based Railroad Non-Stop Customs Clearance System

**DOI:** 10.3390/s23062914

**Published:** 2023-03-08

**Authors:** Sungbeen Kim, Dohoon Kim

**Affiliations:** Department of Computer Science, Kyonggi University, Suwon-si 16227, Republic of Korea

**Keywords:** blockchain, railroad, customs clearance, non-stop, attack resilience, sequence diagram, integrity

## Abstract

Current railroad customs clearance systems are problematic in that the movement of trains is occasionally restricted for extended periods during inspections to verify cargo integrity at customs clearance. Consequently, significant human and material resources are consumed to obtain customs clearance to the destination, considering different processes exist for cross-border trade. Therefore, we developed a cross-border blockchain-based non-stop customs clearance (NSCC) system to address these delays and reduce resource consumption for cross-border trains. The integrity, stability, and traceability of blockchain technology are used to construct a stable and reliable customs clearance system to address these problems. The proposed method connects diverse trade and customs clearance agreements in a single blockchain network, which ensures integrity and minimal resource consumption, and includes railroads, freight vehicles, and transit stations in addition to the current customs clearance system. The integrity and confidentiality of customs clearance data are protected using sequence diagrams and the blockchain to strengthen the resilience of the NSCC process against attacks; the blockchain-based NSCC system structurally verifies the attack resilience based on matching sequences. The results confirm that the blockchain-based NSCC system is time- and cost-efficient compared with the current customs clearance system and offers improved attack resilience.

## 1. Introduction

Cross-border freight transport by rail needs to undergo customs clearance at the border. The clearance process is time-consuming and requires considerable human and material resources. Furthermore, the rail freight agreements between different countries differ from each other [1], with several different consultative bodies responsible for handling the consultation, data integrity, and border delay problems associated with customs clearance processes. In this study, we developed a blockchain-based [2,3,4] border non-stop customs clearance (NSCC) system as a solution to these issues. Trains and freight cars need equipment to operate in a network because NSCC is required to ensure the integrity of goods. In this study, we refer to a setting in which the Internet of Things (IoT) is implemented on trains and freight vehicles [5,6]. Specifically, the cross-border NSCC system overlays a blockchain network on top of the current customs clearance system and uses the combined system as a blockchain platform [2,3,4] within the IoT context. Blockchain technology can address the issue of potential data breaches in cross-border customs clearance because it offers integrity, reliability, and traceability, while also providing the ability to track freight and trains for customs clearance purposes and ensure the reliability of data [7,8,9]. A suitable consensus algorithm for the border NSCC system developed in this study was selected based on previous research [10,11]. A basic attack scenario based on MITRE ATT&CK [12,13,14] was also built to create the NSCC system attack and defense functionality using blockchain technology. Consequently, measures for safeguarding the integrity and confidentiality of the NSCC process were developed, with cyberattacks to which the NSCC is vulnerable actively pre-empted. The structural benefits of NSCC are demonstrated by developing blockchain-based attack resilience [15] and a sequence diagram [16].

The existing customs clearance process transmits customs clearance data using the server–client method, and human resources directly validate cargo integrity. However, NSCC transmits data in a P2P system, with cargo integrity easily checked using the hash function. In addition, compared to the existing customs clearance system, which requires trains to stop at stations, NSCC improves efficiency because it inspects cargo integrity while the train is underway, making this operation the first example of NSCC procedure using blockchain for international rail freight transportation. In addition to resolving the problems associated with the current customs clearance system, the issue of data integrity while the data are being transferred from customs clearance can also be structurally resolved. Furthermore, a blockchain can be effectively applied to other domains because the developed blockchain-based NSCC system offers resilience to common attack scenarios and structural advantages. By utilizing blockchain-based NSCC, a plan was proposed to ensure cyber resilience against attacks targeting station and train nodes that participated in existing customs clearance procedures. In this study, we experimented with the cyber resilience environment that blockchain structurally provides and applied it to existing customs procedures, which enabled us to confirm the structural advantages of blockchain-based NSCC by constructing attack–response scenarios.

The remainder of this article is organized as follows: The structural resilience and consensus process of a blockchain is briefly described in Section 2. The consensus algorithm selected for utilization in this study is discussed, and MITRE ATT&CK is described to structurally prove the attack resilience of the NSCC system. Section 3 presents the designed blockchain-based border NSCC system, and Section 4 discusses the use of a sequence diagram to demonstrate the attack resilience of NSCC, which is the main topic of this study. Section 5 provides the experimental results of NSCC using Simulation of Urban Mobility (SUMO) [17], Docker, and Ethereum [18]. A summary table that explains the current customs clearance system and blockchain-based NSCC system is also presented. Finally, a discussion and the conclusion are presented in Section 6 and Section 7, respectively.

## 2. Related Work

In this study, we develop a blockchain-based NSCC system and demonstrate its structural attack resilience using a blockchain, its consensus algorithm, and an attack sequence diagram created with MITRE ATT&CK. In this section, we discuss relevant prior studies and present a method that applies to this study.

### 2.1. Structural Attack Resilience of Blockchain

In this section, we describe the structural resilience of a blockchain, including the structural benefits of using a blockchain with the NSCC system, as well as the background of this study. A blockchain is a distributed ledger technology in which every node connected to the blockchain network owns the same ledger and is structurally resilient to data forgery, denial of service (DoS) [19], and availability attacks. Consequently, if data forging occurs at one node, accessing another node allows the original ledger to be restored, and data remain preserved until no further nodes remain. Specifically, data robustness can be maintained even if only one node participates in the blockchain network [20].

One drawback of blockchain technology is that it cannot be utilized in various applications currently in operation in the industry. At present, efforts are ongoing in numerous domains to introduce blockchain technology into conventional industries; however, introducing a blockchain structurally is problematic. Incorporating a blockchain in conventional industries is time-consuming because of the high initial introduction cost. Moreover, an attack can target a blockchain network while it is transmitting, receiving, and distributing data when blockchain technology and older systems are combined. However, in this study, we assume that trains, freight cars, and customs clearance stations comprise one IoT node environment. Moreover, we utilize the structural advantages of a blockchain to demonstrate resilience to various attacks targeting industrial networks. In addition, we identify methods to utilize blockchain technology in various industrial domains.

### 2.2. Comparison of Consensus Algorithms and Their Application Domains

In this study, we develop a blockchain-based NSCC system. An integrated higher-level decision-making process is necessary because the NSCC system is a new consensus body created by the collaboration of numerous consensus bodies. To choose the best blockchain consensus algorithm for NSCC, we compare four primary consensus algorithms: proof of work (PoW), proof of activity (PoA), proof of stake (PoS), and delegated PoS (DPoS). Table 1 lists the existing consensus algorithms identified by referring to articles that compared and analyzed several consensus techniques. Table 1 presents PoS, chosen as the consensus technique to be implemented in the proposed blockchain-based NSCC. One drawback of PoW is that it is challenging to introduce in different nations owing to the significant reliance of NSCC on hardware. Additionally, PoA and DPoS are consensus algorithms investigated to compensate for the drawbacks of PoW and PoS, although they are less scalable than PoS. PoS with high scalability is more appropriate for NSCC because it requires the participation of numerous nations and councils. PoS must be required by some validation nodes on the blockchain network. Hence, each node that participates in NSCC must be a validation node, e.g., nations and councils.

To construct a blockchain-based NSCC system, this study chooses the PoS consensus algorithm and obtains consensus from each nation and consultative body. Railway cooperation organizations, such as the Organization for Cooperation of Railways (OSJD) [34] and the Organization for International Carriage by Rail (OTIF) [35], also have freight transport agreements, known as the Agreement on the International Goods Transport by Rail and the Uniform Rules Concerning the Contract of International Carriage of Goods by Rail [36]. The consensus algorithm of a blockchain is similar to a cargo transportation agreement because it is a mechanism that moves forward with customs clearance by evaluating the interests of each country. All consensus algorithms were introduced in various application domains except the railroad industry. Furthermore, because a blockchain uses a consensus algorithm to make decisions, an attack directed at the current network can occur while disseminating the consensus results, instead of the consensus algorithm. Therefore, in this study, we develop a strategy to introduce PoS into the railroad industry, structurally construct a blockchain-based NSCC system, and combine many railroad cooperation groups, such as the OSJD and OTIF, into one consensus system. The NSCC attack–response sequence for a network and cyber threats outlined in Section 2.3 are defined and detailed in Section 4.

### 2.3. Using MITRE ATT&CK

The MITRE organization created MITRE ATT&CK in response to the expansion of the influence of and harm inflicted by cross-border cyberattacks [12,13,14]. The adversarial tactics, techniques, and common knowledge (ATT&CK) framework is a phase of the cyber kill chain model [37] internally designed and arranged based on actual attack cases in MITRE. MITRE ATT&CK is a database consisting of standard data produced by analyzing the adversarial behaviors of attackers from the standpoint of attack tactics and techniques, achieved by observing actual cyberattacks and then classifying and cataloging the attack techniques of various attack groups. MITRE ATT&CK is the result of patterning threatening tactics and techniques to improve the detection of intelligent attacks, taking a slightly different perspective from the traditional cyber kill chain concept. At MITRE, development of the ATT&CK framework initially began by recording tactics, techniques, and procedures (TTPs) relating to hacking attacks employed in the Windows corporate network environment before evolving into a framework that can recognize the behavior of an attacker by mapping TTP information based on studying consistent attack behavior patterns generated by attackers.

In this study, we build a blockchain-based NSCC system attack and defense scenario and demonstrate its attack resilience by applying attack scenarios and sequences created using matrices of MITRE ATT&CK. In conventional security studies, the attack life cycle is depicted as a sequence diagram. We demonstrate the attack resilience of the NSCC developed in this study using this attack sequence diagram. Attack sequence diagrams are frequently used in academic research to support and demonstrate the reliability of networks and systems. By employing a DoS attack against the voice-over-internet protocol [38] environment, for instance, the robustness of the environment, in terms of availability to provide services, can be demonstrated [39].

In this study, an attack sequence diagram is established and used for process analysis to demonstrate the attack resilience of the developed blockchain-based NSCC system. A sequence is constructed in accordance with the basic NSCC process, while the attack life cycle is developed by selecting a random attack point in the sequence.

## 3. Non-Stop Customs Clearance Using Blockchain

The border NSCC system operates in areas in the vicinity of stations located on the borders between different countries. The system is implemented by configuring the blockchain network and enabling data transmission between trains and transit stations that belong to different networks. As shown in Figure 1, a train travels from country A to country B, with network interworking between base stations a and b assumed to be automatic in this process. Go-Ethereum (Geth) blockchain network interworking, required for the border NSCC system to progress, uses a Docker container [40] in the machine of every transit station and train node. Geth software is needed to function as the Ethereum node in the Ethereum network [41]. Considering the Simulation of Urban Mobility (SUMO) framework can simulate the actual traffic environment, it was used to simulate the role of the train in this study. SUMO runs inside the Docker container of the train, with the customs clearance process conducted using communication linking the IP address and port number between Docker containers.

The process developed in this study enables trains to proceed through customs clearance without stopping between transit stations; in addition, the attack resilience is structurally demonstrated. To proceed through Station 2 in country B, a train that already passed through Station 1 communicates the information required for NSCC. After receiving it, Station 2 checks the integrity of the data by comparing it to the hash value [42,43] of the data stored in the current blockchain network. The hash value for the cargo specs is broadcast to the blockchain network and other transit stations during the departure process once the cargo is loaded on the train at the original departure point. Cargo integrity is examined by contrasting the hash value transmitted with the hash value of the data propagated throughout the process of passing through each transit station. The hash value of the data is compared with the hash value recorded on the blockchain, and if no discrepancies are found, the NSCC process continues. If a problem arises, the train and its cargo are inspected using the existing customs clearance process.

The operation of the NSCC system is described in Section 3.1, with the network setup needed to use the blockchain defined and explained in Section 3.2.

### 3.1. Procedure of Blockchain-Based Non-Stop Customs Clearance System

The NSCC process is divided into five individual steps, as shown in Figure 2, where each step is described. NSCC uses distributed storage as its data storage process because it utilizes a blockchain network [44]. Data are logged using the distributed storage system known as the interplanetary file system [45,46,47], and data comparison and verification are conducted. Data are recorded and stored using distributed storage, a network of distributed nodes. Consequently, every train node and transit station node involved in the blockchain network participates in the distributed storage system. For the comparison–verification process, the customs clearance data are uploaded to the distributed storage and encrypted using the hash function. The respective steps of the process are shown in Figure 2.

**(Step 1)** Enter and transact: A train node approaches a station node by this process to conduct the NSCC process. The customs clearance data (raw data) are processed by the train node and sent to the station node for customs clearance. This process employs a security network (e.g., a virtual private network) that utilizes the base station of each country [48,49].**(Step 2)** Receive and hash: Data from the train node are relayed to the station node, which hashes the data using a hash function. The calculated hash value is compared and validated in Step 3. The hash function to be used at this point is chosen from SHA-256 [50] or Keccak-256 [51] and applied throughout the customs clearance process.**(Step 3)** Compare: The station node compares the hash value of the hashed data with that of the initial customs clearance data generated when the cargo was initially loaded. The hash value uploaded to the distributed storage is currently compared with that produced by the station node based on the transaction recorded in the blockchain. The results of the comparison are broadcast in Step 4.**(Step 4)** Broadcast: The success of the NSCC process is determined by comparing the hash value produced by the station node to that in the distributed storage. Subsequently, the train node decides on whether to proceed. If the hash value of the distributed storage that already exists differs from that generated in the relevant station node, the train proceeds in accordance with the existing customs clearance procedures. If the two hash values correspond, indicating that no irregularities exist with the data or cargo, the train node passes through without stopping. The passing information is broadcast to other stations and train nodes.**(Step 5)** Dashboard: A dashboard displays the NSCC-related data. The visualized data can be examined and subsequently analyzed. The corresponding dashboard of each node allows users to view information about the blockchain network and hardware resources.

Potential attack points for each node, component, interface, and layer that constitute NSCC are shown in Figure 2 and Figure 3. Points A, B, and C represent potential weak points vulnerable to attacks, which can be attacked by hostile attacker nodes intending to damage the NSCC network and systems. Attacks on NSCC-related data and communications are possible through these points. In this study, we use the properties of the blockchain to structurally demonstrate the attack resilience of points A, B, and C. In Figure 3, target, network, and storage are the three potential attack layers, with an attack scenario created by setting an attack sequence diagram of the respective points. The data shown in Figure 3 can be breached and stolen by the blockchain-based NSCC based on the configured scenario.

For instance, if the station in Figure 2 is attacked, data relevant to customs clearance can be compromised, making the customs data verification process vulnerable to attacks. In the event of an attack, significant issues, such as time delays and misjudgment can occur during the customs clearance process. Moreover, sensitive data can be compromised because the customs clearance process follows an international consensus procedure. However, the attack resilience of the customs clearance node is structurally proven by the developed NSCC sequence diagram, with the method for securing resilience explained using security elements as an example.

### 3.2. Network Configuration of Blockchain-Based Non-Stop Customs Clearance System

This section describes the organizational structure of the blockchain network, and the manner data are sent to and received from the network via an existing railway network. The structure of the blockchain-based NSCC network is depicted in Figure 4, with the NSCC sequence technique from the perspective of each node summarized in Table 2. The detailed explanation is as follows:

The network for NSCC based on a blockchain, shown schematically in Figure 4, indicates that each IP address uses the same subnet mask because when setting up the experimental environment, the network is configured utilizing many Docker containers inside a single machine. The IP information of each node is expressed differently during the actual NSCC application process. For example, Stations A and B have static IP addresses of 242.42.25.65 and 103.132.54.12, respectively. Furthermore, the port number increases sequentially, as shown in Figure 4; however, when NSCC is applied, appropriate port numbers, such as 9090 and 7897, can be assigned to each node.

The number of connected nodes is also changed if NSCC is applied to customs clearance. The network includes the customs clearance nodes from 29 OSJD and 51 OTIF member countries, assuming that the present customs clearance offices in border areas are participating (as of December 2022) [52]. As more member nodes join the blockchain network, it becomes more stable. Therefore, the NSCC network has high robustness, and the maturity of each node increases with the number of consultative bodies and countries participating in NSCC.

A summary of each process depicted in Figure 4 is provided in Table 2, as identified by the process number. This process is more difficult than the current railroad customs clearance system because data broadcast from a train to a transit station uses network connection protocols and interfaces. An attacker can target the network, trains, and transit stations in this process. Sequence diagrams are used in this study to describe the basic flow of this process. In addition, each attack–defense phase is defined and the structural resilience of the blockchain-based NSCC system is demonstrated.

## 4. Attack Resilience in Blockchain-Based Railway

A sequence diagram for basic customs clearance is defined and systematically discussed in Section 4.1. In addition, the potential attack time and method, which are the focal elements of this study, are presented in Section 4.2, where the attack–response sequence diagram is defined. The procedures for attack, response, and analysis are described. The attack–response sequence diagram constructed based on potential attack points A, B, and C is depicted in Figure 2 and Figure 3.

### 4.1. Basic Sequence of Blockchain-Based Non-Stop Customs Clearance

The basic flow of the developed blockchain-based NSCC is shown in Figure 5. When the initial cargo information is transmitted to the blockchain network at a shipping point, the transaction status and block in the blockchain are returned. Once the cargo is recorded in the blockchain network, the train departs for the transit station. When the train passes through this transit station, data pertaining to customs clearance are broadcast to the station and verified. The verification process corroborates the integrity of the cargo based on the data already recorded in the blockchain network. A hash function is used to readily and rapidly substantiate the customs data, as shown in Figure 2.

The sequence diagram shown in Figure 5 only depicts the basic customs clearance process of NSCC, with additional details of each step omitted for clarity. Furthermore, all ‘Station’ mentioned below are categorized as customs clearance station nodes according to Figure 5. As the diagram is intended to indicate the steps involved in the automatic generation of data recorded throughout the transaction and block creation process of each blockchain and the verification process at each transit station, the processes related to distributed storage are not shown here.

Given that a blockchain is a platform that offers integrity and reliability, the NSCC process can be conducted by relying on these features of a blockchain for customs clearance in trains and transit stations. As presented in Section 4.2, an attack–response sequence is added to the basic NSCC sequence to reinforce the structural robustness of the NSCC and ensure the cyber resilience of the blockchain-based NSCC.

### 4.2. Attack Sequence of Blockchain-Based Non-Stop Customs Clearance with Attack Resilience

The attack–response sequence diagram for the potential attack points of the blockchain-based NSCC system, which is the main concept of this research, is presented in this section. The entire sequence demonstrates that the blockchain structurally has attack resilience, with an attacker performing an attack sequence against a victim and the victim responding in correspondence with the sequence. As defined in Figure 5, a customs clearance station node is referred to as a Station.

#### 4.2.1. Attack Sequence A: Attacking Clearance Station Node Using DoS

Blockchain-based NSCC technology is robust and has attack resilience in terms of the availability of customs clearance. In this sequence, an attacker targets Station A with a DoS [19] attack. We utilize the structural benefits of the blockchain to defend the system against this attack.

**Attack sequence:** One of the potential attack points indicated in Figure 2 and Figure 3 is transit Station A, through which the train is expected to pass. An attacker prepares a DoS attack against this target. In addition to overloading the network communication of transit Station A by packet fragmentation, the attacker sends a request to establish socket communication to transit Station A [53]. Accordingly, the train waits at the station without transmitting a request to pass through after receiving data relating to customs clearance and completing the verification process. Thus, the attacker keeps transit Station A overloaded to perform DoS attacks and delay customs clearance.**Corresponding sequence:** As the train node cannot receive permission to pass through transit Station A, it sends a request to other nearby transit station nodes for customs clearance. When transit Station B receives a request for customs clearance, the train is granted permission to pass through Station B, and the train is processed for passage through transit Station A in accordance with the existing customs clearance sequence.**Analysis and discussion:** A flowchart based on the attack–response scenarios for DoS attacks is shown in Figure 6. As all trains and transit station nodes are connected to the blockchain network, no problems occur when clearance is requested from and processed by transit Station B. If the integrity of the data transmitted from the train can be verified, the train can pass through the customs clearance station without stopping. Thus, the system is designed to enable other trusted customs clearance nodes to handle the data verification process. The attacker targets the availability of the NSCC; however, it offers resilience against these attacks because only one blockchain network is used. As the blockchain network is structurally designed to ensure the reliability and integrity of recorded transactions, even if additional nodes participate in the verification process, the reliability of the verification is ensured.

#### 4.2.2. Attack Sequence B: Attacking Distributed Storage Using Spoofing Attack

In terms of the data integrity and reliability of customs clearance, the developed blockchain-based NSCC is attack-resilient and robust. In this sequence, an attacker conducts a spoofing attack [54] targeting the distributed storage. The structural benefits of the blockchain are utilized to defend against this attack.

**Attack Sequence:** Figure 7 shows the approach followed to target and attack the distributed storage system of the blockchain-based NSCC. An attacker confounds the sender by spoofing the domain address and routing details to connect to the distributed storage at B, a potential attack point, as depicted in Figure 2 and Figure 3. Customs documents are sent to transit Station A by train. Transit Station A utilizes the distributed storage and transaction data on the blockchain network to verify them. During this process, the attacker transfers arbitrary data while changing the routing table of transit Station A to enable the attacker to appear as the distributed storage. Data inconsistency occurs because transit Station A undertakes the verification process based on the data sent by the attacker; consequently, the NSCC process cannot be implemented.**Corresponding sequence:** Transit Station A analyzes the distributed storage data for inconsistencies and compares them to transactions [55] on its own local blockchain ledger. After the first verification, a secondary verification is conducted with the transaction data of the actual blockchain network because the hash value of the customs clearance data is present in the transaction data. Transit Station A updates the routing table and broadcasts permission for the train to pass through after verifying that the data from the blockchain network correspond with the customs clearance data. The sequence is completed after the customs clearance is recorded on the blockchain network.**Analysis and discussion:** An attack that targets the routing database occurs when a transit station is proceeding with verification. The distributed storage and blockchain network transactions contain the data needed for verification, and any data inconsistencies can be determined in the event of an attack directed against the distributed storage. In this case, the blockchain network is accessed to verify data because every participating node has the same ledger. The blockchain platform structurally ensures integrity, reliability, and traceability because all participating nodes share the same ledger. These features of the blockchain can be used to safely conduct the data verification process.

#### 4.2.3. Attack Sequence C: Attacking Clearance Station Nodes Using Advanced Persistent Threat and Backdoor Attacks

In terms of data integrity and customs clearance resilience, NSCC based on blockchain technology is attack-resilient and robust. In this sequence, an attacker targets Station A using advanced persistent threat (APT) [56] and backdoor [57] attacks. These attacks are warded off by utilizing the structural benefits of the blockchain.

**Attack sequence:** The attacker is based at potential attack point C, as shown in Figure 2 and Figure 3. The process before the attack is the same as the basic NSCC process. However, when a train departs, the attacker designates the transit station along the route as a target, launches an APT attack, and simultaneously inserts a backdoor. If the attack is successful, the attacker can control the root authority of transit Station A [58] and modify the transaction data of the blockchain. Subsequently, a discrepancy arises between the data transmitted and received by the train during the verification of customs clearance data with transit Station A.**Corresponding sequence:** The train that is refused customs clearance sends its request for permission to pass through to nearby transit Station B. The customs clearance data are checked at transit Station B, which responds with the necessary permission for customs clearance. Furthermore, data sync to transit Station B is requested to restore the blockchain transaction data of transit Station A, which is falsified. To recover the transaction data of transit Station A and conduct its ledger sync process, transit Station B and other transit stations transfer the entire blockchain data to transit Station A, which can re-participate in the customs clearance process.**Analysis and discussion:** Root access can be hijacked using numerous methods. Figure 8 shows a straightforward example of backdoor injection via an APT attack. A transit station with social engineering issues is vulnerable to root authority hijacking attacks. This attack falsifies the blockchain data of a transit station node and interferes with customs clearance. Owing to the structural features of the blockchain, data can be restored even if the blockchain data inside one node are altered. All nodes included in the blockchain can participate in the consensus process, as shown in Figure 6. Consequently, transit Station B is required to continue with customs clearance.

## 5. Experimental Results

In this section, we describe the simulation of the environment in Figure 1 using SUMO, Docker, and Ethereum for experiments, explain the blockchain-based NSCC, and demonstrate the advantages of the approach followed in this study. In addition, the conventional concept of a blockchain is explained, and the quantitative and qualitative metrics used when a blockchain is integrated into the customs clearance system are presented. Table 3 provides information about the software versions and status information of the experimental environment.

### 5.1. Experiments and Materials

In this study, the NSCC environment was constructed using SUMO, Docker, and Ethereum, with the final experimental results of the NSCC derived based on the experimental results of each component. SUMO makes it possible to simulate the navigation of a given road network by single vehicles in response to a given traffic demand. The simulation is purely microscopic: each vehicle is modeled explicitly, has its own route, and navigates the network individually. Therefore, we used SUMO to derive the travel route and the timing of the train on the railroad. In addition, NSCC was implemented using Docker to configure the train and station as one node, while Geth was used to implement the blockchain network. Figure 9 shows the structure of our experimental environment.

Station and Train nodes are implemented as a single Docker container; considering they are implemented on a single local machine for experimentation, they all have the same IP address and different port numbers. However, in a real environment, all nodes have different IP addresses and port numbers. Each node was composed of Geth nodes to connect to the Ethereum network; in the case of the Train node, the SUMO client was run inside the Docker container to serve as a train in this experiment. Each Docker container communicates using the IP address and port number. During the communication process, the Docker container transmits and receives data, interlocks with the Ethereum network, and propagates transactions and blocks. In this study, using SUMO, the NSCC was tested for a simulation involving trains moving between Kazakhstan and Mongolia. Table 4 shows the values and parameters required for the experiment. The train departed from Station 1, a customs clearance station in Mongolia, and traveled to Station 2, a customs clearance station in Kazakhstan, at approximately 150 km/h.

Figure 10 shows the SUMO-based simulation environment, with Kazakhstan to the left of the red line and Mongolia to the right. We simulated a train traveling from Station 1 in Mongolia to Station 2 in Kazakhstan. Specific information is listed in Table 4, with the train movement and NSCC procedures tested in this environment. The steps indicated below the figure correspond to Steps 1–4 described in Figure 2. Step 0 involves the train moving from Station 1 to Station 2. While the train is underway, in Step 1, data related to customs clearance are transmitted to the station. In Step 2, the station proceeds with hashing based on the received data. In Step 3, the station compares and verifies the data integrity based on the hash value of the data. In Step 4, the station propagates the final verification result to other station and train nodes. The novelty of this study is that after reducing the speed to a minimum from Step 1 to Step 4, the train passes through the station. In the case of the existing customs clearance procedure, it is necessary to stop at the station to allow the cargo to be inspected. In contrast, the proposed basic methodology for our blockchain-based NSCC enables the train to pass through the station without stopping after cargo integrity is verified.

### 5.2. Results of Blockchain-Based NSCC

This study developed a solution to the problem of time delays caused by the need to obtain customs clearance, consumption of human and material resources, and reliability and integrity of customs clearance of the existing customs clearance system. The system utilizes a Bitcoin-derived blockchain [22]. A blockchain is a peer-to-peer-based system in which distributed nodes share a single ledger [59], and is characterized by integrity, reliability, and traceability. Data integrity can be realized by alerting other nodes when data are forged because nodes are distributed and share the same ledger. Additionally, data are recorded in a setting that ensures integrity. The reliability of previously created data and blocks increases if data are transmitted because transactions and blocks are continuously created by a verification process. Owing to their integrity and reliability, data continually entered into the ledger on a blockchain cannot be falsified. Therefore, the traceability of all data can also be guaranteed.

The average processing times for Step 1 to Step 4 are listed in Table 5. The amount of data allowed per transmission in the process is limited to a maximum of 1000 MB. In Step 1, data are propagated using a VPN environment, which takes a maximum of 10 min based on the minimum data propagation speed over the network. Step 2 takes a maximum of 1 min based on the minimum execution time of SHA-256, which is 20 Mbps in size, to derive a hash value using the SHA-256 hash function after receiving the data. Step 3 takes a maximum of 10 min based on the minimum download speed of 1 Mbps from IPFS to perform the comparison verification based on the data stored in IPFS. Finally, when the verification process is completed based on the customs clearance data, it takes a maximum of 10 min to verify transactions and propagate blocks using the PoW consensus algorithm in the Ethereum-based PoW environment, calculated at 11 TPS. Therefore, the overall process takes a maximum of 31 min to complete if no problems arise. However, considering that connection to the blockchain may involve delays or disconnections depending on the network environment, a maximum of 1 h is considered necessary to enable these issues to be resolved [60].

Using these features of a blockchain network and software, we created a “blockchain-based border NSCC” system in this study and tested its resilience against attacks. The main contributions of this study and the differences between the developed NSCC and current customs clearance systems are discussed below.

Table 6 compares the traditional customs clearance system with the blockchain-based NSCC system. The six criteria used for comparison are time, resource, integrity, reliability, transparency, and traceability. The respective criteria are described as follows:**Time:** The current customs clearance process is time-consuming because individuals have to directly inspect customs clearance items and cargo. However, with NSCC, customs clearance can be completed in as little as 1 h if the validity of the customs documents is not questionable.**Resource:** In the current customs clearance system, people directly participate in customs clearance and personally inspect the goods and cargo. However, resource consumption is minimal because the accuracy of the customs data is verified by machine. Customs clearance is conducted by verifying the integrity using the hash value of the data, which is broadcast to the blockchain network.**Integrity:** Data integrity is safeguarded by the distributed ledger technology used in the blockchain. However, data forgery and tampering can occur because documents are stored in a database and written by hand in the current customs clearance system.**Reliability:** The current customs clearance system assumes that the people participating in customs clearance are reliable. However, the blockchain-based NSCC system can structurally ensure reliability.**Transparency:** The blockchain-based NSCC guarantees that the customs clearance process remains transparent. The participation of each of the member countries in verification and customs clearance enables transparent data management. However, the transparency of the current customs clearance process cannot be ensured because of possible threats by malicious attackers.**Traceability:** The current customs system tracks data to documents and databases. However, the blockchain-based NSCC uses a distributed storage and blockchain network to track every step of the continuous customs clearance process from shipment to unloading.

The comparison in Table 6 shows that blockchain-based NSCC guarantees integrity, reliability, and traceability, reduces the need for human resources, and shortens the time required for customs clearance. In addition, our solution based on the consensus algorithm of the blockchain integrates the interests of existing railway cooperation agreements and organizations, such as OSJD and OTIF. However, the blockchain-based NSCC system is still vulnerable to APT and network attacks aimed at legacy systems, which prompted us to consider the attack resilience of NSCC, as demonstrated in Section 4, using sequence diagrams to provide a structural explanation. Section 6 and Section 7 outline the limitations and future research directions.

## 6. Discussions

In this study, we developed a blockchain-based NSCC system intended as a new customs clearance mechanism with structural robustness and attack resilience. As demonstrated in Section 4, existing customs clearance systems are vulnerable to DoS, APT, and spoofing attacks. The proposed blockchain-based NSCC system includes a method to solve the above-mentioned problems based on the integrity and reliability provided by the blockchain. In addition, the efficiency of the customs clearance process was maximized by reducing the time required for the customs clearance procedure to 1 h. However, real measurements are challenging and significant system resources are required to implement the sequences, as described in Section 4. In the future, each of these sequences can be investigated and additional vulnerabilities to cyberattacks could be considered, with attack sequences tested by modeling and simulation (M&S) [65,66,67]. In this study, the current railroad customs clearance system was set up as an overlay network for the developed blockchain-based NSCC system. The integrity and reliability offered by a blockchain can be ensured when configured as an overlay. However, because each train and transit station node participates as nodes in the blockchain network, machine resources unnecessary in the conventional customs clearance system are required. Our choice of a PoS-based consensus algorithm minimizes the use of computational resources. We plan to perform M&S for each potential consensus method in our next study.

This study was conducted based on an IoT-based network environment. Further research on IoT and artificial intelligence (AI)-based block seals and smart container capabilities [68] for inspecting cargo integrity is required, which we plan to incorporate in future studies. Concepts of IoT-based block seals and AI can be used to actively inspect cargo integrity in terms of damage and movement. This study is the first step toward proposing a blockchain-based customs clearance procedure. The most important aspect of this study is that we introduced blockchain into the existing railway customs clearance process to maximize the efficiency of the international railway customs clearance process.

## 7. Conclusions

This study involved developing a blockchain-based NSCC system and structurally demonstrating its resilience to cyberattacks. Cyberattacks aimed at legacy systems can occur because the NSCC system combines a blockchain with the legacy customs clearance system. However, an attack–response sequence diagram was used to demonstrate that cyberattack resilience can be secured by employing the integrity, reliability, and traceability features of the blockchain.

Compared to the current customs clearance methods, the blockchain-based NSCC system excels in terms of integrity, security, and reliability. Reducing the time required for customs clearance can improve the performance of the freight transportation sector using railroads for cross-border trade. Consequently, the developed customs clearance method uses fewer materials and people overall. This study demonstrated the versatility of blockchain technology and its implications for maritime and aviation trade and the customs clearance system for cross-border railroad transport.

This work demonstrated the compatibility of blockchain with traditional systems. Future research could employ trade domains, such as land, sea, and air. The blockchain-based NSCC system proposed in this study can also be improved using IoT and AI-based object recognition systems to verify cargo integrity. Furthermore, M&S of the NSCC system can be conducted based on the created attack–response sequence diagram presented in Section 4 to appropriately apply the environment, such as the consensus algorithm and network protocol. The defense system utilizing MITRE D3FEND can also be extended [69]. Moreover, simulating the connection between the actual train model and IoT equipment based on the experiment conducted in Section 5 could be explored further.

## Figures and Tables

**Figure 1 sensors-23-02914-f001:**
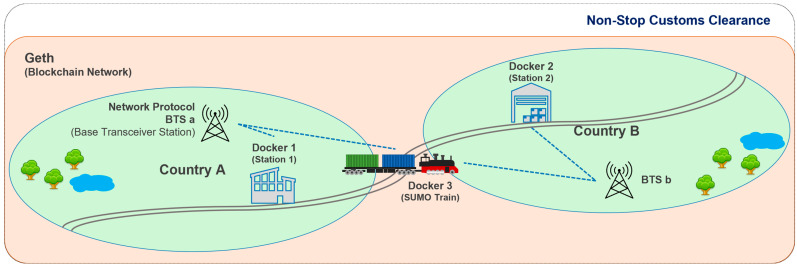
Railway border non-stop custom clearance in real-world simulation.

**Figure 2 sensors-23-02914-f002:**
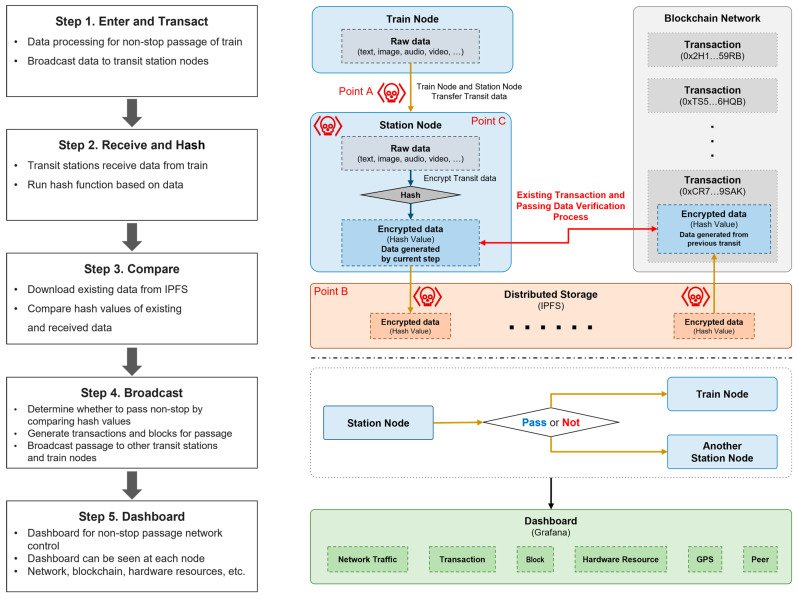
Process overview of non-stop customs clearance system and attack points.

**Figure 3 sensors-23-02914-f003:**
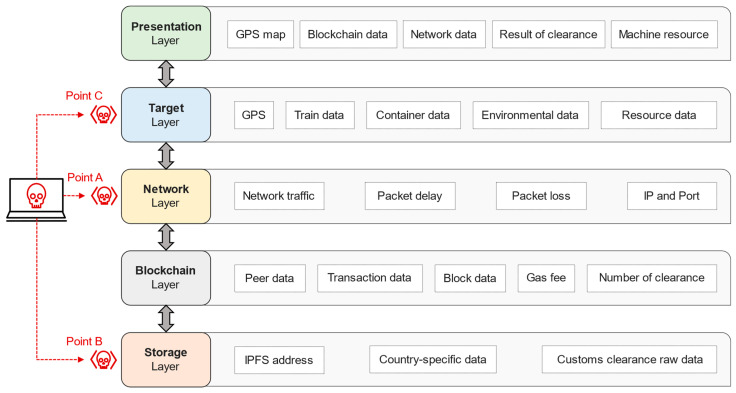
Data derived for each component layer.

**Figure 4 sensors-23-02914-f004:**
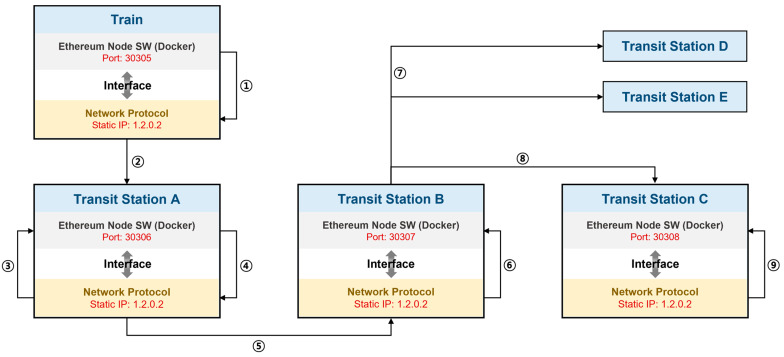
Non-stop customs clearance network implemented with Ethereum.

**Figure 5 sensors-23-02914-f005:**
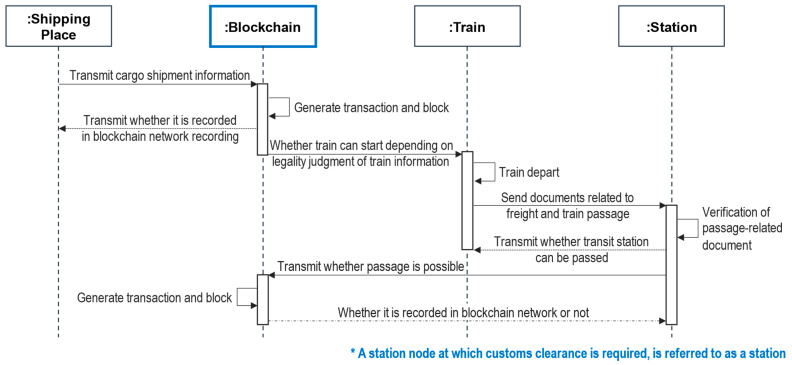
Basic sequence diagram of NSCC.

**Figure 6 sensors-23-02914-f006:**
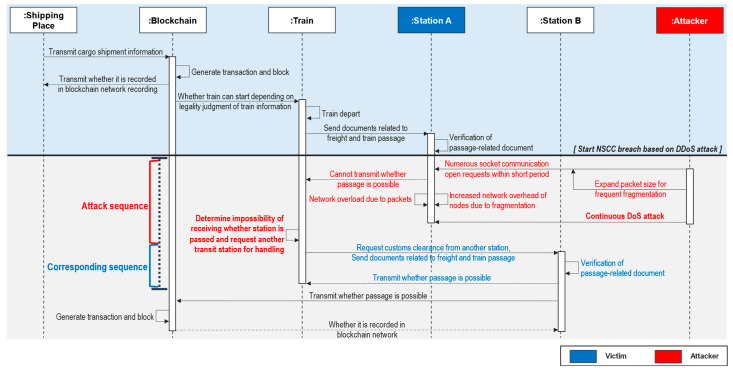
Attack sequence diagram using DoS.

**Figure 7 sensors-23-02914-f007:**
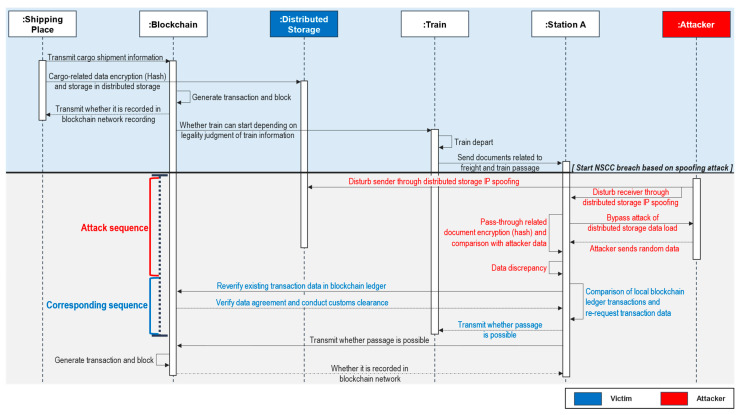
Attack sequence diagram using spoofing.

**Figure 8 sensors-23-02914-f008:**
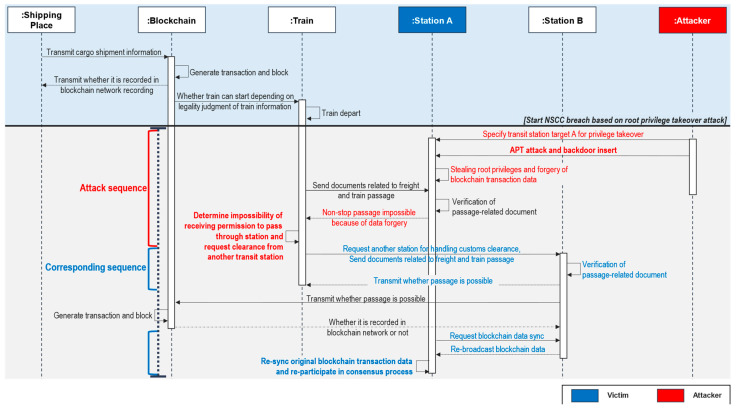
Attack sequence using root access privileges.

**Figure 9 sensors-23-02914-f009:**
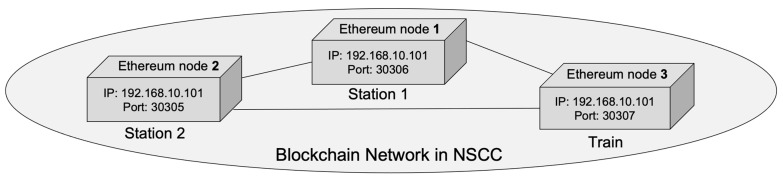
Blockchain network in NSCC using Docker, SUMO, and Ethereum.

**Figure 10 sensors-23-02914-f010:**
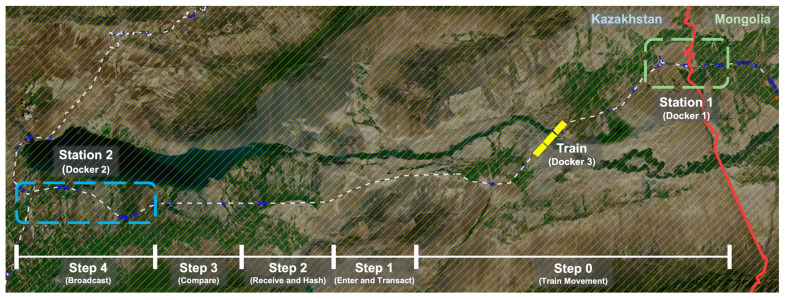
Simulation of NSCC from Mongolia to Kazakhstan using SUMO.

**Table 1 sensors-23-02914-t001:** Quantitative indicator analysis of consensus algorithms [11,21].

Category	PoW	PoA	PoS	DPoS
Latency (response time)	10 min	5 min	1 min	3 s
transaction per second (TPS)	≥7 TPS	≥14 TPS	≥300 TPS	≥500 TPS
Computing overhead	High	Low	Medium	Medium
Scalability	Low	Medium	High	Medium
Decentralized level	High	Low	Medium	Medium
Hardware dependency	Yes	No	No	No
Security (in application)	Low	Medium	Medium	Medium
Consensus method	Hash rates	Activity-based	Stake	Stake votes
Reference	[22,23,24]	[25,26,27]	[28,29,30,31]	[32,33]
**Adequacy**	**X**	**△**	**O**	**△**

**Table 2 sensors-23-02914-t002:** Process of non-stop customs clearance on blockchain network.

Process No.	Description of Each Process
①	Communication data cleaning and communication protocols are accessed to transmit data from trains to transit stations.
②	Data transmitted to transit station using communication protocol of machine.
③	Transit station that received data through communication protocol accesses Ethereum node to verify data.
④	After verification process, communication protocol to deliver data to another transit station is accessed.
⑤	Verified results broadcast to other transit stations (blockchain network) using communication protocol.
⑥	Verification data received from other transit stations through blockchain network are checked.
⑦	Data forwarded to other transit station nodes that do not directly participate in this customs clearance process, and data are verified.
⑧	Verification-related data are transmitted to transit stations on future train routes.
⑨	Transit stations other than those that received data check whether transaction information of blockchain network matches the verification result.

**Table 3 sensors-23-02914-t003:** Simulation and experimental environment.

Category	Description
OS	Windows 11
GPU	RTX 3070 Ti
RAM	16 GB
Docker OS	Ubuntu 20.04
Blockchain environment	Geth v1.10.25
SUMO version	SUMO v1.14.1

**Table 4 sensors-23-02914-t004:** Numerical values and parameters required for experiments.

Category	Descriptions of Values and Parameters
Stations and country	Mongolia Station 1 to Kazakhstan Station 2
Train velocity	Approximately 150 km/h
Coordinates of departure station	lat: 44.162919, lon: 80.326560
Coordinates of destination station	lat: 43.632262, lon: 77.647001
Maximum duration of consensus algorithm	Up to 10 min with PoW

**Table 5 sensors-23-02914-t005:** Processing speed and available data size for each NSCC step.

Category	Time for Each Procedure	Speed	Data	Ref.
Step 1: Enter and transact	Up to 10 min with VPN	5~10 Mbps	1000 MB	[61]
Step 2: Receive and hash	Up to 1 min with SHA-256	20 Mbps	1000 MB	[62,63]
Step 3: Compare	Up to 10 min with IPFS	1 Mbps	1000 MB	[46]
Step 4: Broadcast	Up to 10 min with PoW	11 TPS	.	[64]
**Total**	Up to 31 min	.	.	.

**Table 6 sensors-23-02914-t006:** Quantitative comparison of the existing customs clearance system and NSCC.

Category	Existing Customs Clearance	Non-Stop Customs Clearance (Ours)
Time to customs clearance	About 1–2 days	Up to 1 h (from Table 5)
Resource	Human and machine	Machine
Integrity	X	O
Reliability	△	O
Transparency	△	O
Traceability	X	O

## Data Availability

Not applicable.

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
