# Peer review of "Securing the Cyber Resilience of a Blockchain-Based Railroad Non-Stop Customs Clearance System"

_sensors, 2023, doi:10.3390/s23062914_

Round 1
Reviewer 1 Report
1. What are the contributions of the paper?
2. What are the merits compared to the existing approach?
3. Research design should be appropriate.
4. English language and style are fine/minor spell check is required.
Author Response
Please review the attached file.

Reviewer 2 Report
This paper presents a method for securing the cyber resilience of a blockchain-based railroad non-stop customs clearance system.
- I would instead add a new section, experimental results, to emphasize the exact content in the section.
- Please perform an experimental comparison with some other recently published results.
- This will help justify whether the proposed method has advanced the state-of-the-art in this field.
- It is essential to improve the quality and all figures and make all detailed texts readable.
Author Response
Please review the attached file.

Reviewer 3 Report
1. The authors must emphasise its contribution clearly.
2. It is of the utmost importance to examine the research gaps and why secure cyber resilience of blockchain must be studied.
3. This study's introduction requires a compelling narrative to convey its originality.
4. Table 1 may contain the conclusion from the quantitative indicator analysis of consensus algorithms.
5. Please discuss Figure 1 and determine which portion has been explored in prior research. Does prior research address how SUMO 3 has interacted with docker 1 and docker 2?
6. Include the constraint as well as the implications for theory and practise. What is the most important aspect in terms of sensor journal reader benefits?
Author Response
Please review the attached file.
